General retrieval network model for multi-class plant leaf diseases based on hashing

Yang Zhanpeng 1
Wu Jun wjglo@huat.edu.cn 2 3
Yuan Xianju 20160001@huat.edu.cn 1
Chen Yaxiong 4
Guo Yanxin 1
1 School of Automotive Engineering, Hubei University of Automotive Technology , Shiyan , China
2 School of Mathematics, Physics and Optical Engineering, Hubei University of Automotive Technology , Shiyan , China
3 Hubei Key Laboratory of Applied Mathematics, Hubei University , Wuhan , China
4 School of Computer Science and Artificial Intelligence, Wuhan University of Technology , Wuhan , China
Chaki Jyotismita
Electronic publication date: 2024 Nov 26
Publication date: 2024
Volume: 10
Electronic Location ID: e2545
Received 2024 Jul 10; Accepted 2024 Nov 5
Copyright: ©2024 Yang et al.
Copyright year: 2024
Copyright holder: Yang et al.
License: This is an open access article distributed under the terms of the Creative Commons Attribution License, which permits unrestricted use, distribution, reproduction and adaptation in any medium and for any purpose provided that it is properly attributed. For attribution, the original author(s), title, publication source (PeerJ Computer Science) and either DOI or URL of the article must be cited.
License URL: https://creativecommons.org/licenses/by/4.0/

Keywords: Plant disease, Retrieval, Hashing learning, Convolutional neural network, Deep learning

Funding: Natural Science Foundation of Hubei Province 2022CFB959 Educational Commission of Hubei Province of China Q20221802 Hubei Key Laboratory of Applied Mathematics HBAM202105 Doctoral Fund of Hubei University of Automotive Technology BK202114 This work was supported by the Natural Science Foundation of Hubei Province (Grant No. 2022CFB959), the Educational Commission of Hubei Province of China (Grant No. Q20221802), the Hubei Key Laboratory of Applied Mathematics (Grant No. HBAM202105) and the Doctoral Fund of Hubei University of Automotive Technology (Grant No. BK202114). The funders had no role in study design, data collection and analysis, decision to publish, or preparation of the manuscript.

==============================
Traditional disease retrieval and localization for plant leaves typically demand substantial human resources and time. In this study, an intelligent approach utilizing deep hash convolutional neural networks (DHCNN) is presented to address these challenges and enhance retrieval performance. By integrating a collision-resistant hashing technique, this method demonstrates an improved ability to distinguish highly similar disease features, achieving over 98.4% in both precision and true positive rate (TPR) for single-plant disease retrieval on crops like apple, corn and tomato. For multi-plant disease retrieval, the approach further achieves impressive Precision of 99.5%, TPR of 99.6% and F-score of 99.58% on the augmented PlantVillage dataset, confirming its robustness in handling diverse plant diseases. This method ensures precise disease retrieval in demanding conditions, whether for single or multiple plant scenarios.

Introduction

Plants play a critical role in sustaining human life and their health is crucial for agricultural productivity and ecological balance. However, traditional methods of plant disease and pest identification or prediction, such as visual inspection and field trapping, though widely used in agricultural practice, have become increasingly limited. These methods are often time-consuming, prone to human error and may lead to delayed responses to disease outbreaks, particularly in large-scale agricultural operations. Additionally, the vast variety of plant diseases and their high similarity present significant challenges in accurate identification. The lack of large-scale, efficient retrieval models further exacerbates the difficulties in managing and predicting plant diseases in modern ecological agriculture.

In recent years, as agriculture has gradually moved towards more sustainable and ecological practices, modern agricultural technologies based on machine learning have been introduced for the identification (Putzu, Di Ruberto & Fenu, 2016; Li, Gao & Shen, 2010) classification (Le et al., 2020; Campanile, Di Ruberto & Loddo, 2019) and retrieval (Kebapci, Yanikoglu & Unal, 2010; Bama et al., 2011; Piao et al., 2017; Baquero et al., 2014) of plant diseases. For instance, machine learning models have improved detection efficiency and reduced human intervention through automated data processing. However, these methods still face challenges related to complex feature extraction. One major issue is that it is necessary to design imaging schemes based on the differences in plant diseases For example, Gurjar (2011) proposed a method based on the varying color and shape features of different cotton diseases, suggesting that distinct feature extraction schemes are necessary for each disease type. Similarly, selecting appropriate lighting conditions is crucial for achieving high-quality image analysis. Putzu, Di Ruberto & Fenu (2016) pointed out that many proposed methods are only effective under controlled lighting conditions and uniform backgrounds. To address this, Putzu developed a mobile application for leaf analysis, which automatically identifies plant species and solves the primary problem of uncontrolled lighting with highly accurate results. Furthermore, environmental factors such as meteorological conditions also play a significant role. Yun et al. (2015) proposed a method that integrates the statistical features of leaf images and meteorological data. By incorporating features like color, texture and shape from images taken in various environments, alongside meteorological characteristics, their approach tackles the influence of changing weather conditions on machine learning-based plant disease detection models. While carefully designed imaging schemes can significantly reduce the difficulty of creating classical algorithms, they come at the cost of increased application complexity. Additionally, in natural environments, classical algorithms struggle to avoid the impact of environmental changes on model performance. This emphasizes the need for more adaptive and robust solutions in real-world agricultural settings.

To overcome these limitations in traditional methods, deep learning techniques have been introduced into the field of plant disease detection and classification. By leveraging the powerful feature extraction capabilities of deep learning, the accuracy of plant disease detection (Shougang et al., 2020; Luo et al., 2023; Nain, Mittal & Hanmandlu, 2024; Verma, Kumar & Singh, 2023; Peyal et al., 2023) and classification (Kaya & Ercan, 2023; Ulutaş & Veysel, 2023) has been significantly improved. Several deep learning-based models have shown significant potential in real-world agricultural applications. For instance, Zhu et al. (2021) introduced an innovative method for carrot appearance detection, utilizing convolutional neural networks (CNNs) to extract image features and support vector machines (SVMs) for classification. This approach achieved an accuracy of 98% in classifying carrot appearance quality, offering significant potential for enhancing the efficiency of automated carrot sorting systems. Similarly, Fang et al. (2020) developed a CNN-based model for identifying varying degrees of disease severity across different crops. This model provides valuable insights for disease diagnosis and the precise application of pesticides, contributing to the reduction of pesticide misuse through accurate assessment of disease severity. However, the application of traditional deep learning models in large-scale plant disease image retrieval still faces challenges, particularly in efficiently and accurately retrieving images of multiple diseases with high similarity. Moreover, high-precision retrieval models for multiple plants and their diseases remain scarce in the current literature.

In the field of image retrieval, hashing techniques are renowned for their efficient retrieval speed, compact storage and strong performance in approximating similarities and have been widely applied in large-scale image retrieval (Gui et al., 2018; Xiaoqiang, Xiangtao & Xuelong, 2017; Lu, Chen & Li, 2018). Hashing techniques convert high-dimensional data into compact binary codes, which accelerates the retrieval process and reduces storage requirements while maintaining high accuracy. Additionally, a key advantage of hashing is its collision-resistant property, which allows it to effectively distinguish between highly similar features and subtle differences.

Despite these strengths, hashing techniques have not been fully explored in agricultural disease retrieval. This paper addresses this gap by introducing the deep hash convolutional neural network (DHCNN) into large-scale plant leaf disease retrieval. The DHCNN leverages hashing’s ability to manage high similarity and fine-grained differences among plant diseases—capabilities that traditional methods may struggle with. Initially, the research will focus on retrieving single plant diseases, with plans to expand to the retrieval of multiple plant diseases. This approach aims to develop a comprehensive multi-plant leaf disease retrieval model and compare its performance with traditional methods.

This study contributes to the existing large-scale plant leaf disease retrieval literature in the following ways:

1. The DHCNN model is introduced into the field of large-scale plant leaf disease retrieval, offering a novel deep learning-based approach aimed at improving retrieval performance in this domain.

2. A detailed comparison was conducted between the performance of DHCNN and both traditional machine learning-based plant disease retrieval models and more recent deep learning-based models, highlighting the model’s potential advantages.

3. To validate the robustness and scalability of the proposed model, experiments were performed on three widely recognized plant disease datasets—PlantVillage, PlantDoc, and the Plant Disease Dataset. Additionally, ablation studies were conducted to ensure the model’s reliability.

4. By leveraging the collision-resistant property of hashing techniques, this approach is designed to enhance retrieval accuracy for multiple plant leaf diseases in large-scale image retrieval tasks, demonstrating the advantages of this technique in improving retrieval performance.

The remainder of the paper is organized as follows: the “Related Work” section discusses the application of machine learning and deep learning in the field of plant disease detection. The “Proposed Methodology” section provides a detailed explanation of the working principles of the proposed model and the implementation of the retrieval process. In the “Experiments” section, plant disease detection datasets are introduced, along with the evaluation metrics and parameter settings applied in the model. The “Results and Analysis” section presents a performance analysis of the model and compares it with other models. Additionally, the “Conclusion” summarizes the main findings of our work. The “Future Directions” section outlines potential research focuses and development directions.

Related Work

In this section, an analysis of the literature on plant disease identification, classification, and retrieval is provided. The methods proposed by researchers for plant disease identification, classification and retrieval are categorized into two types: machine learning methods and deep learning methods.

Machine learning approaches in plant disease

Putzu, Di Ruberto & Fenu (2016) proposed a mobile application for leaf analysis and automatic plant species identification. This application primarily addresses the identification and segmentation steps, tackling the major issue of uncontrolled lighting conditions and achieved highly accurate results. Li, Gao & Shen (2010) introduced a method for the automatic identification of three wheat diseases by analyzing morphological features extracted from images. This method segments images of wheat powdery mildew, wheat sheath blight and wheat stripe rust to obtain target areas, extracts and optimizes morphological data and combines principal component analysis with discriminant analysis. The method ultimately selects sphericity, roundness, Hu1, Hu2 and equivalent radius as recognition factors, achieving sample identification rates of 96.7%, 93.3% and 86.7% for the three wheat diseases, respectively. Yun et al. (2015) considered the statistical features of leaf images and meteorological data, proposing a crop disease identification method that successfully identified three cucumber diseases with an accuracy of 90%. Le et al. (2020) proposed a novel method that improves the discrimination rate for broadleaf plants by combining features extracted using local binary pattern operators with those from plant leaf contour masks. This method filters noise in plant images using morphological opening and closing operations and classifies crops and weeds using a support vector machine classifier. It achieved a classification accuracy of 98.63% on the “bccr-segset” dataset, significantly outperforming previous methods with an accuracy of 91.85%. Campanile, Di Ruberto & Loddo (2019) developed an open-source ImageJ plugin for image analysis in the biological field, particularly focusing on botany research. This plugin extracts morphological, texture and color features from seed images for classification purposes. Yağ & Altan (2022) developed a hybrid model using FPA, SVM and a simplified CNN for efficient plant disease detection. By selecting optimal features from 2D-DWT transforms of plant images, the model achieves high accuracy with low computational complexity. Deployed on an NVIDIA Jetson Nano, it enables real-time classification of plant diseases, offering a fast and lightweight solution for agricultural applications. Kebapci, Yanikoglu & Unal (2010) proposed a method combining color, shape and texture features for plant image retrieval. They first used the max-flow min-cut (MFMC) graph cut method for interactive segmentation to separate plants from the background. By extracting color, shape and texture features and inputting them into a SVM classifier, effective plant image retrieval was achieved. This research considered not only the local features of plant leaves but also the overall shape, background and flower color of the plants, providing a comprehensive feature description for plant image recognition. Bama et al. (2011) proposed a plant leaf image retrieval method based on shape, color and texture features. They first converted RGB images to the HSV color space and applied Log Gabor wavelets to the saturation channel to extract texture features. They then used the scale-invariant feature transform (SIFT) algorithm to extract key points and performed corner detection for key point elimination. Finally, descriptor ratio matching was used for leaf image retrieval. This method effectively improved the accuracy of plant leaf image retrieval, especially for images with complex backgrounds. Piao et al. (2017) proposed a method combining image descriptors to enhance the performance of crop disease image retrieval systems. By first calculating the similarity between images using a single descriptor and then summing the similarities of multiple descriptors to generate a new similarity score, the system completed image retrieval based on this. Experimental results showed that the use of combined descriptors universally improved system performance, with different optimal descriptor combinations found for different crops. Baquero et al. (2014) proposed an image retrieval strategy based on color structure descriptors and nearest neighbor algorithms for diagnosing greenhouse tomato leaf diseases. This strategy successfully characterized various anomalies such as chlorosis, sooty mold and early blight, aiding non-experts in assessing and classifying tomato disease symptoms. Shrivastava, Singh & Hooda (2016) proposed an efficient soybean leaf disease detection and classification method based on image retrieval. The study introduced two feature descriptors, HIST and WDH and explored various color and texture feature descriptors, such as BIC, CCV, CDH and LBP, for automated soybean disease recognition. The method effectively detected and classified disease regions using background subtraction techniques and validated its robustness in segmenting six soybean diseases.

From the discussion of machine learning methods above, these methods, while simple to apply, require large amounts of training data and heavily rely on manual expertise. Moreover, these techniques are not robust to the wide variability in the size, color and shape of plant diseases. Therefore, newer methods are needed to improve the accuracy of identifying multiple plant leaf diseases.

Deep learning approaches in plant disease

To overcome these limitations of machine learning methods in agriculture, deep learning has been gradually applied in this field  (Fuentes et al., 2017). Shougang et al. (2020) proposed a deconvolution-guided VGG network (DGVGGNet) model to address issues such as shadows, occlusions and varying light intensities in plant leaf disease recognition. This model uses VGGNet for disease classification training, recovers classification results as feature maps using deconvolution and combines upsampling and convolution operations to achieve deconvolution. Finally, it performs binary classification training for each pixel with minimal disease patch supervision, guiding the encoder to focus on true disease areas. Experimental results showed that this model performed exceptionally well in complex environments, with a disease type recognition accuracy of 99.19%, pixel accuracy for lesion segmentation of 94.66% and an average intersection over union (IoU) score of 75.36%. Luo et al. (2023) utilized FPGA to accelerate convolutional neural networks (CNNs) for plant disease recognition, addressing memory and computational constraints on edge devices. They designed a compact seven-layer network, “LiteCNN”, with 176K parameters and 78.47M FLOPs. After optimization and validation on the ZYNQ Z7-Lite 7020 FPGA board, the model achieved an accuracy of 95.71%, showcasing a low-power, high-accuracy and fast solution suitable for real-time plant disease detection. Nain, Mittal & Hanmandlu (2024) explored various color space models to enhance plant disease detection, finding that the HSL-CNN joint model achieved the highest accuracy of 98.97%, precision of 98.61% and F1-score of 99.28%. Verma, Kumar & Singh (2023) proposed a single lightweight CNN model for detecting diseases in maize, rice and wheat, achieving accuracies of 99.74%, 82.67% and 97.5%, respectively. This model outperformed multiple benchmark CNNs with only 387,340 parameters, making it suitable for real-time use in resource-constrained environments. Peyal et al. (2023) developed a lightweight 2D CNN architecture for early detection of diseases in cotton and tomato crops, achieving an average accuracy of 97.36%, outperforming pre-trained models such as VGG16, VGG19 and InceptionV3. This model was implemented in the “Plant Disease Classifier” Android application for efficient, smartphone-assisted plant disease diagnosis. Sharma & Sharma (2024) introduced a semi-supervised and ensemble learning framework to improve plant disease diagnosis, addressing issues like dataset scarcity and annotation costs. The framework achieved 18.03% higher accuracy and a 15% boost in F-score, with a 13.35% improvement in mean average precision, offering practical benefits for sustainable agriculture. Kaya & Ercan (2023) introduced a new deep learning-based plant disease detection method by integrating RGB and segmented images for disease identification. The method employed a multi-head DenseNet architecture and was evaluated on the PlantVillage public dataset, which contains 38 classes and a total of 54,183 images. The model achieved an average accuracy of 98.17%, recall of 98.17%, precision of 98.16% and F1-score of 98.12% in five-fold cross-validation, demonstrating high robustness and low standard deviation, indicating effective differentiation of plant diseases with varying features. Ulutaş & Veysel (2023) developed an integrated deep learning model for early detection of nine tomato leaf diseases, achieving a high classification accuracy of 99.60%. Their approach combined two newly proposed CNN models with well-known architectures, such as MobileNetV3Small and EfficientNetV2L, using particle swarm optimization and grid search for fine-tuning and hyperparameter optimization. This research highlights the efficiency and speed of these models in diagnosing plant diseases, providing valuable tools for preventing infection spread. Guowei et al. (2024) introduced DFN-PSAN, a deep feature fusion network for plant disease classification in natural environments. By combining YOLOv5 with pyramidal squeezed attention (PSA), the model enhances key image regions and improves accuracy. Tested on three datasets, DFN-PSAN achieved over 95.27% accuracy, with PSA reducing parameters by 26%. t-SNE and SHAP were used to improve model interpretability. Dubey & Dilip (2024) developed a machine learning-based method for plant disease detection. Key features are extracted from pre-processed images and an optimized artificial neural network (ANN), enhanced by the adaptive sunflower optimization (ASFO) algorithm, classifies diseases. The model achieved 97.94% accuracy, demonstrating its effectiveness for early plant disease diagnosis. Chen et al. (2020) introduced a novel deep learning architecture, INC-VGGN, for plant disease image recognition, with a validation accuracy of 91.83% on a public dataset. Even under complex background conditions, the model achieved an average accuracy of 92.00% for rice disease image classification. Argüeso et al. (2020) proposed a Few-Shot Learning (FSL) approach for plant leaf classification, addressing classification issues in small datasets. This study used the Inception V3 network for fine-tuning the source domain and transferred the learned knowledge to the target domain, achieving nearly 90% accuracy with very few training samples using Siamese networks and Triplet loss. Experimental results demonstrated that this FSL method significantly improved classification performance on small datasets compared to traditional fine-tuning transfer learning methods. Loddo, Loddo & Di Ruberto (2021) proposed SeedNet, a novel convolutional neural network for seed classification and retrieval tasks. Through deep learning methods, they achieved satisfactory performance in seed retrieval tasks, providing strong support for developing comprehensive seed recognition, classification and retrieval systems. Peng & Wang (2022) introduced a novel image retrieval system for automatic detection, localization and identification of plant leaf diseases. By optimizing the YOLOv5 algorithm, this system improved recognition capabilities for small objects and combined classification with metric learning to flexibly identify new disease types without retraining. Detailed experiments on three public leaf disease datasets demonstrated the system’s effectiveness in quickly determining disease types.

The analysis of existing crop disease identification, classification and retrieval methods is summarized in Table 1. From Table 1, it is evident that deep learning still has gaps in the field of multi-plant disease retrieval.

Table 1 The Summary of existing crop disease identification, classification, and retrieval methods.

	Data type	Task type	Result	Evaluation metrics	
Li, Gao & Shen (2010)	Wheat disease images	Identification	96.7%, 93.3%,86.7%	Accuracy	
Gurjar (2011)	Plant leaf images, meteorological data	Recognition	90%	Accuracy	
Le et al. (2020)	Crops and weeds images	Classification	98.63%	Accuracy	
Campanile, Di Ruberto & Loddo (2019)	Seed images	Classification	99.1% accuracy with combined features	Accuracy	
Kebapci, Yanikoglu & Unal (2010)	Plant leaf images	Content-Based Image Retrieval	Precision: 0.6, Recall: 0.62	Precision, Recall	
Bama et al. (2011)	Crop disease images	Image retrieval	–	F-measure	
Piao et al. (2017)	Tomato leaf images	Image retrieval	–	Precision, Recall, F-measure	
Yun et al. (2015)	Tomato plant disease and pest images	Recognition	99.19%	Accuracy	
Verma, Kumar & Singh (2023)	Cotton and tomato image	Recognition	97.36%	Accuracy	
Nain, Mittal & Hanmandlu (2024)	Disease and healthy leaf images of corn, rice, and wheat	Multi-class plant disease recognition	Accuracy: 98.97%, Precision: 98.61% F1-score:99.28%	Accuracy, Recall, Precision, F1-Score	
Peyal et al. (2023)	Plant-Village dataset	Multi-plant classification	Accuracy: 98.17% Recall: 98.17% Precision: 98.16% F1-score: 98.12%	Accuracy, Precision, Recall, F-measure	
Shrivastava, Singh & Hooda (2016)	Rice disease images	Classification	92.00%	Accuracy	
Fuentes et al. (2017)	Plant leaf images	Classification	90%	Accuracy	
Guowei et al. (2024)	Seed images	Classification retrieval	–	–	

Current status and limitation

Despite the advancements in plant disease identification, classification and retrieval, as discussed above, there is still a notable gap in the development of models specifically designed for multi-plant disease retrieval. While models have been successfully developed for the recognition and classification of multiple plant diseases, as well as for the retrieval of plant seeds and individual plant diseases, the field lacks robust solutions for multi-plant disease retrieval. This gap highlights the need for a comprehensive approach that can handle the complexity and variability of diseases across different plant species. To address this, our study introduces a novel multi-plant disease retrieval method based on hash learning, which aims to bridge this gap and provide an efficient solution for large-scale plant disease retrieval.

Proposed Methodology

In recent years, a deep hash neural network (DHNN) was proposed for large-scale remote sensing image retrieval (RSIR) (Li et al., 2018), demonstrating the potential of deep hashing techniques in complex scenarios. Additionally, Song, Li & Benediktsson (2021) demonstrated the effectiveness of deep hash CNNs in handling complex image retrieval tasks involving large datasets.

In the context of plant disease retrieval, many plant diseases and pests exhibit strong similarities in appearance, such as the shape, extent and location of lesions, which can lead to misjudgments with traditional methods. For instance, the curling of leaves due to plant diseases or pests can appear very similar across different conditions and stages. Figure 1 provides several sets of images showing similar disease characteristics. The high levels of similarity between features in these cases often lead to misclassification by conventional networks. Hashing techniques, however, are particularly adept at identifying subtle differences within feature distributions due to their collision-resistant properties. Even minor differences in the encoded features can result in significant distinctions after hashing. This characteristic makes hashing methods particularly effective for distinguishing between similar plant diseases and pests, thus improving retrieval accuracy.

Figure 1 Pairs of plant leaf images with similar lesion characteristics.

Building on this foundation, a novel disease retrieval method for plant leaves based on DHCNN is proposed. This approach is particularly well-suited to handle the challenges posed by leaf images, which often exhibit significant intra-class variance and limited inter-class variance. The DHCNN approach is designed to minimize the feature distance between similar image pairs while maximizing it for dissimilar pairs. By leveraging paired inputs, DHCNN learns the similarity and dissimilarity information between images, extracts discriminative semantic features for accurate classification and generates compact hash codes for efficient retrieval. The DHCNN architecture includes a pre-trained CNN, a hash layer and a fully connected layer with a softmax classifier, providing a robust solution for plant disease retrieval.

In the implementation, a pre-trained CNN is employed within DHCNN to initially extract deep features. Subsequently, the hash layer transforms these high-dimensional deep features into low-dimensional hash codes using metric learning regularization. Furthermore, the fully connected layer, combined with the softmax classifier, is utilized to generate the image class distribution. Based on the hash codes and class distribution, the retrieval process employs hash sorting to identify images similar to the given input image. More detailed information about DHCNN will be provided in ‘Deep Feature Extraction’ and ‘Hash-based Metric Learning and Loss Function’ as the paper progresses.

Deep feature extraction

The training of a CNN requires a large number of samples. Moreover, annotating unknown specimens is costly and time-consuming. In order to deal with these obstacles, a pre-trained CNN model for reducing requirements for extensive training samples is introduced in an instance.

In this study, the VGG11 (Simonyan & Zisserman, 2014) model was chosen as the backbone network for the deep feature extraction part, primarily due to its simplicity and strong generalization capabilities. The VGG network, known for its straightforward architecture with eight convolutional layers and three fully connected layers, provides a good balance between depth and computational efficiency, making it highly suitable for a wide range of image retrieval and classification tasks. Moreover, the model’s architecture, characterized by the simplicity of its convolutional layers and fully connected layers. The consistent structure of VGG11 facilitates transfer learning, allowing the model to adapt well to different datasets with minimal fine-tuning. Additionally, its use of smaller convolutional filters (3x3) helps in capturing fine details in the images, which is essential for distinguishing between similar plant diseases.

The specific process is illustrated in Fig. 2. The “maxpool” operation characterizes the size of max pooling, “4096” represents the feature dimension for the fully connected layers and the dropout technique is applied to the fully connected layers 9 and 10. The activation function for all weight layers is the rectified linear unit (ReLU) (Krizhevsky, Sutskever & Hinton, 2017).

Figure 2 Feature extraction process.

A pre-trained CNN is introduced for the extraction of deep features and a hash layer is added after the fully connected layer to convert the high-dimensional deep features into compact K-bit hash codes.

It is supposed that there are N training samples, denoting as (1a) X=xii=1N.

The corresponding set of labels can be represented as (1b) Y=yii=1N

where yi ∈ ℝC is the ground-truth vector of sample xi with only one element of 1 and others are 0, C is the total number of image scene classes. For arbitrary image Xi ∈ X , we can extract its deep features (i.e.,the output of the fully connected layer in Fig. 2.), giving by (1c) fi=Φxi;θ,i=1,2,…,N

in which Φ is the network function characterizing by θ which denotes all the parameters of the VGG.

Hash-based metric learning and loss function

Paired inputs are utilized by DHCNN for training the network and the process will be illustrated as that of Fig. 2.

Specifically, xi,xj represents a pair of images and symbol Sij is defined for it. Usually, Sij is 1 if xi and xj are from the same class. Otherwise, it is 0. deep features fi,fj are obtained by forward propagation in this instance. Subsequently, a hash layer will be connected after the fully connected layer in this model, as shown in Figs. 2 and 3A. The high-dimensional feature vector from the fully connected layer is transformed into hash-like features through a linear transformation and these hash-like features are then mapped to the hash space using the sign function, so that the high-dimensional deep features can be converted into compact K-bit hash codes, which will be followed that (Song, Li & Benediktsson, 2021) (2) ut=Whft+Vh

where ut is the class hash feature, Wh ∈ ℝK×4096 denotes the weight matrix, Vh ∈ ℝK×1 represents the bias vector, ft ∈ ℝ4096 represents the high-dimensional feature vector obtained after processing through the deep convolutional neural network.

Figure 3 Hash layer and softmax classifier in deep feature transformation.

The hash layer incorporating metric learning regularization is employed to transform high-dimensional deep features into low-dimensional hash codes (A). The specific details can be found in Eqs. (2), (3) and (4). Furthermore, a fully connected layer equipped with a softmax classifier is employed to produce the distribution of classes (B).

The function signx is a sign function used to convert real numbers into binary hash codes. Specifically: (3) signx=1,ifx≥0−1,ifx<0

(4) bt=signut,t=i,j.

In the case where the hash code B=btt=1N is known. The likelihood function of paired labels S=sij can be defined by Song, Li & Benediktsson (2021) (5) psij∣B=φωij,sij=11−φωij,sij=0

in which φ(⋅) is the logistic function, φx=11+e−x and ωij=12biTbj . According to above definitions, the loss function is derived by taking the negative log-likelihood of the observed pairs of labels in S. Discretely reformulating the loss function gets (Song, Li & Benediktsson, 2021): (6) L1=−∑sij∈Ssijψij− log1+eψij+β∑i=1Nui−bi22

where ψij=12uiTuj,i,j=1,2,…,N and β is the regularization parameter that constrains ui to converge to bi . By minimizing L1, the Hamming null distance between similar samples can be optimized to the minimum. Conversely, the Hamming distance between different samples is maximized.

Unlike image retrieval methods based solely on the similarity information between images to learn hash codes, DHCNN incorporates semantic information from each image to enhance the feature representation capability (Li et al., 2018). The specific process will be demonstrated in Fig. 3B. In detail, a fully connected layer with a softmax function after the hashing layer will be inserted for generating a category distribution for each image. Additionally, the cross-entropy loss is used to minimize the error between the predicted labels and the ground truth labels, following that De Boer et al. (2005) (7) L2=−1N∑i=1Nyi,logsoftmaxWsuk+vs

where Ws ∈ ℝC×K and vs ∈ ℝC×1 denote the weight matrix and bias vector, respectively. Minimizing the loss function L2, the CNN can be adapted to learn the discriminative semantic features of each image.

As mentioned as above, L1 can be utilized to acquire the similarity information between images and L2 is intended to learn the label information of each image. To this end, a new loss function is defined for improving the network performance by considering both similarity information and label information, yielding (8) L3=ηL1+1−ηL2

where η ∈ [0, 1] is the regularization parameter that balances the label information and similarity information. The label information of each image is only considered if η is 0 and the similarity information between images is required if η is equal to 1. Finally, the objective function is the minimum loss function L3 , which is given by (9) J= minL3= minηL1+1−ηL2.

Solving the problem in Eq. (7), the stochastic gradient descent (SGD) algorithm is also considered to learn the parameters.

Completing DHCNN training, the hash codes and class labels of all samples from the database are tried to obtain. For any image xq , its hash code bq and class label cq can be determined by (10) bq=sgnuq=sgnWhfq+vh

(11) cq=argmaxk=1,2,…,Ctik

where tik is the kth component of vector ti. Eventually, a query image is the input. The DHCNN model can quickly and accurately output a set of similar images after sorting the Hamming distance between the query image and the images in the database as well as the class distribution. The retrieval process is shown in Fig. 4.

Figure 4 Schematic diagram of retrieval based on hash codes and class distribution.

Experiments

Datasets and data augmentation

For retrieval training, a publicly available dataset of plant disease images called Plant-Village (Geetharamani & Pandian, 2019) is selected. This dataset has 39 different categories of plant leaf and background images, generally including 38 categories of plant leaf disease and totaling 54,305 images. There is also a separate category containing 1,143 background images.

Various image augmentation techniques were employed to increase the number of images in the dataset and avoid overfitting. These techniques include image flipping, gamma correction, noise injection, PCA color enhancement, rotation and scale transformation. The strategy adopted for data augmentation was to augment the categories with fewer than 1,000 images to 1,000 images, while leaving the categories with 1,000 or more images unchanged. Subsequently, 1,000 images were selected from each augmented category. For the training and testing phases, a distribution percentage of 80% for training and 20% for testing was adopted. This 80/20 split was chosen based on standard practices in the field, which balance the need for a robust training set with the necessity of having a representative test set to evaluate model performance accurately. The choice ensures that the model can learn effectively from a sufficient number of samples while still being tested on a diverse subset of data. Additionally, multiple trials with different random splits were performed to verify the stability and reliability of the results. The final dataset consists of 7,600 images in the test set and 30,400 images in the training set. Detailed parameters of the dataset are shown in Table 2.

Table 2 Categories of leaf disease datasets.

Class name	Number of images	
	Without using data augmentation	Using data augmentation	Training set	Test set	
Apple with scab	630	1000	800	200	
Apple with black rot	621	1000	800	200	
Apple with cedar apple rust	275	1000	800	200	
Healthy apple	1645	1645	800	200	
Blueberry with healthy	1502	1502	800	200	
Cherry with healthy	854	1000	800	200	
Cherry with powdery mildew	1052	1052	800	200	
Corn with grey leaf spot	513	1000	800	200	
Corn with common rust	1192	1192	800	200	
Healthy corn	1162	1162	800	200	
Corn with northern leaf blight	985	1000	800	200	
Grape with black rot	1180	1180	800	200	
Grape with black measles	1383	1383	800	200	
Healthy grape	423	1000	800	200	
Grape with leaf blight	1076	1076	800	200	
Orange with Huanglongbing	5507	5507	800	200	
Peach with bacterial spot	2297	2297	800	200	
Healthy peach	360	1000	800	200	
Pepper with bacterial spot	997	1000	800	200	
Healthy pepper	1478	1478	800	200	
Potato with early blight	1000	1000	800	200	
Healthy potato	152	1000	800	200	
Potato with late blight	1000	1000	800	200	
Healthy raspberry	371	1000	800	200	
Healthy soybean	5090	5090	800	200	
Squash with powdery mildew	1835	1835	800	200	
Healthy strawberry	456	1000	800	200	
Strawberry with leaf scorch	1109	1109	800	200	
Tomato with bacterial spot	2127	2127	800	200	
Tomato with early blight	1001	1001	800	200	
Healthy tomato	1591	1591	800	200	
Tomato with leaf mold	952	1000	800	200	
Tomato with septoria leaf spot	1771	1771	800	200	
Tomato with two spotted spider mite	1676	1676	800	200	
Tomato with target spot	1404	1404	800	200	
Tomato with mosaic virus	373	1000	800	200	
Tomato with yellow leaf curl virus	5357	5357	800	200	
Tomato with late blight	1909	1909	800	200	

Evaluation metrics

In experiments, five metrics are used to evaluate the performance of retrieval methods. Of those, precision measures the accuracy of positive predictions. It is defined as the ratio of true positives to the sum of true positives and false positives, yielding (12) Precision=TPTP+FP.

Recall, also known as sensitivity or true positive rate (TPR), is the ratio of true positives (TP) to the sum of TP and false negatives (FN). It measures the model’s ability to correctly identify all positive instances, following that (13) TPR=Recall=TPTP+FN.

False negative rate (FNR) is the ratio of false negatives (FN) to the sum of true positives (TP) and FN. It measures the proportion of actual positive instances that are incorrectly identified as negative by the model. (14) FNR=1−TPR=FNTP+FN.

Mean average precision (MAP) is the mean of the average precision scores for each class in a multi-class classification problem. It evaluates the precision of the model across different recall levels, which is given by (15) MAP=1N∑i=1NAPi

where APi is the average precision (AP) for class i and N is the number of classes.

The F-score is the harmonic mean of precision and recall. It provides a balance between Precision and Recall, particularly when the class distribution is imbalanced. (16) F-score=2×Precision×RecallPrecision+Recall.

Area under the curve (AUC) is the area under the receiver operating characteristic (ROC) curve, which plots the TPR against the FPR. It measures the overall ability of the model to distinguish between positive and negative classes. (17) AUC= ∫01TPRxdx

where TPR(x) is the TPR as a function of the FPR.

Parameter setting

Adopting the Plant-Village dataset, the DHCNN model is trained and optimized by adjusting different hyperparameters for obtaining best results. Tables 3 and 4 shows the structure and optimized hyperparameters for the proposed DHCNN model.

Table 3 The hash retrieval model structure with VGG as the backbone network.

Layer	Configuration	
Conv1	Filter 64 × 3 × 3, stride 1 × 1, pad 1, LRN, maxpool 2 × 2	
Conv2	Filter 128 × 3 × 3, stride 1 × 1, pad 1, maxpool 2 × 2	
Conv3	Filter 256 × 3 × 3, stride 1 × 1, pad 1	
Conv4	Filter 256 × 3 × 3, stride 1 × 1, pad 1, maxpool 2 × 2	
Conv5	Filter 512 × 3 × 3, stride 1 × 1, pad 1	
Conv6	Filter 512 × 3 × 3, stride 1 × 1, pad 1, maxpool 2 × 2	
Conv7	Filter 512 × 3 × 3, stride 1 × 1, pad 1	
Conv8	Filter 512 × 3 × 3, stride 1 × 1, pad 1, maxpool 2 × 2	
Full9	4096, dropout	
Full10	4096, dropout	
Hash1	Hash layer	

Table 4 Training parameters settings.

Parameters	Value	
Model	VGG11	
Batch Size	32	
Optimizer	SGD	
Loss Function	J	
Epoch	200	
Hash Code Lengths	64	
Learning Rate	0.1	
Training set	30400	
Test set	7600	

The epoch is a hyperparameter that defines a single pass through the complete training set during the training of a deep learning model. It refers to the number of times the entire training dataset passes forward and backward through the neural network. In this study, the epoch was set to 200. During experimentation, it was observed that the model’s results began to stabilize around epoch 40. However, the epoch was set to 200 to ensure that the model had ample opportunity to learn from the data and to avoid premature convergence.

The batch size is another crucial hyperparameter that affects computation efficiency, memory usage, gradient estimation accuracy and training stability. In this study, batch sizes of eight, 16, 32, 64, and 128 were experimented with. After thorough investigation, the batch size of 64 was found to provide the best balance between training speed and model performance.

The learning rate is a hyperparameter that controls the step size of the model’s parameter updates during each iteration. It significantly influences both the training speed and the final performance of the model. For this study, an initial learning rate of 0.1 was chosen. To fine-tune the training process, the learning rate was scheduled to decrease by a factor of 0.1 after every 30 epochs. This decay strategy helped maintain the stability of the training process, ensuring that the model converges effectively without overshooting the optimal solution.

Additionally, the hash code length, a critical parameter for the deep hash CNN model, was varied among 16, 32 and 64. The model achieved optimal performance with a hash code length of 64, balancing between retrieval accuracy and computational efficiency.

Results and Analysis

Search results for single plant disease

The individual retrieval results for five plant types apple, corn, grape, potato and tomato are presented in Table 5.

Table 5 Retrieval MAP for each plant.

Plants	Number of classifications	Number of images per classification	Accuracy of this article	Accuracy of others	
		Training set	Test set			
Apple	4	800	200	1	0.905 (Simonyan & Zisserman, 2014)	
Corn	4	800	200	0.984	–	
Grapes	4	800	200	0.997	0.639 (Bama et al., 2011); 0.996 (Simonyan & Zisserman, 2014)	
Potatoes	3	800	200	0.995	0.538 (Krizhevsky, Sutskever & Hinton, 2017)	
Tomato	10	800	200	0.994	0.6 (Piao et al., 2017)	

The convergence of five retrieval models starts at epoch of 50. Besides corn, the accuracy of all models is over 99%. The retrieval of apple plants converges quickly because of distinct differences in features of various disease images and the accuracy approaches 1, which presents the 10% improvement compared to the results achieved by Karthikeyan & Raja (2023). However, corn leaves’ retrieval accuracy is slightly lower than other plants since the characteristic images of corn gray leaf spot and corn rust. Also, northern corn leaf blight are highly similar. Nevertheless, the accuracy still reached 0.984. Owing to the large number of disease types, the retrieval results for tomato plants are relatively lower. However, the average retrieval accuracy still reaches 0.994, which presents the 40% improvement compared to the results achieved by Baquero et al. (2014). The average retrieval accuracy for grape and potato plants reaches 0.997 and 0.995 respectively, which represents the 36% and 46% improvement relative to the results obtained by Piao et al. (2017) and Patil & Kumar (2013). Additionally, the retrieval results for the grape dataset are similar to those obtained by Karthikeyan & Raja (2023).

For establishing models for multiple plants and diseases in practice with high accuracy, the retrieval system results for disease identification in the leaves of multiple plants will be introduced in next sections.

Search results for multiple plant disease

In this section, 38 plant diseases from the Plant-Village dataset will be applied to retrieval experiments for the multi-plant disease. The DHCNN model is adopted to calculate J loss for validating performance of the model. The training loss values varied with different Epoch are shown in Fig. 5.

Figure 5 The training loss of the Plant-Village dataset with different epochs.

In the Epoch-Loss curve, the loss value gradually decreases if the Epoch is increased from 1 to Epoch 40. Such a phenomenon indicates that the model is gradually learning the patterns and features in the dataset. The loss value tends to be stable if the Epoch is larger than 40 and increases gradually. At this point, the model has converged and the training of the retrieval model is completed.

As is shown in Fig. 6, the precision is still higher than 0.995 if the k is small than 800. Here, k denotes the number of top-ranked images retrieved and evaluated. This indicates that the system proposed here with the retrieval ability of extremely high precision and stability. Following the increase of k, the precision will be decreased if it is larger than 800 since most relevant images in the dataset have already been retrieved.

Figure 6 The precision of the DHCNN on a multi-plant dataset at different values of k.

The precision remains above 0.995 when k is less than 800, demonstrating the system’s high precision and stability. However, as k increases beyond 800, precision decreases as most relevant images have already been retrieved.

The recall-k curve is unaffected by imbalanced datasets and effectively reflects the changes of true positive rate and false positive rate of retrieval model at different thresholds, showing as Fig. 7. Clearly, as k is increased from 1 to 800, the recall steadily rises and approaches 1. This trend signifies the retrieval system’s effectiveness in identifying more positive samples while keeping false negatives to a minimum. However, when k exceeds 800, the recall value stabilizes. This indicates that beyond this point, the relevant images in the retrieval dataset have already been effectively retrieved, and further increases in k do not significantly impact the recall value.

Figure 7 The recall of the DHCNN on a multi-plant dataset at different values of k.

As k increases from 1 to 800, recall steadily rises towards 1, indicating effective identification of positive samples with minimal false negatives. Beyond k = 800, recall stabilizes, suggesting that relevant images have been effectively retrieved, and further increases in k do not significantly impact recall.

In order to validate the retrieval performance, a so-called recall-precision curve which characterizes a relationship between recall and precision under different thresholds is shown in Fig. 8. Evidently, the precision on the horizontal axis and recall on the vertical axis illustrates the precision and recall values at different k values. The optimal balance point is found in the top-right corner where both precision and recall tend to be 1. Such a point corresponds to a k value of 800 in the precision-k and recall-k curves. At this point, the model presents the best overall retrieval performance, thus retrieving as many relevant images as possible and minimizing false positives.

Figure 8 The precision–recall curve of the DHCNN on a multi-plant dataset.

The precision-recall curve shows precision and recall values at different k values, with the optimal balance point at the top-right corner where both are close to 1. This point corresponds to a k value of 800 in the curves, indicating the best retrieval performance with maximum relevant images retrieved and minimized false positives.

To evaluate the retrieval performance, Figure 9 presents the ROC curve, showing the relationship between the TPR (recall) and the FPR at various thresholds. The y-axis represents the TPR, while the x-axis shows the FPR. With an AUC of 0.997, the curve highlights the model’s high retrieval accuracy, effectively capturing relevant results while minimizing false positives. The curve’s near approach to the top-left corner reflects a well-optimized balance between true positive retrieval and reducing false positives across different thresholds.

Figure 9 The ROC curve of the DHCNN on multi-plant dataset.

The ROC curve evaluates the model’s retrieval performance. The x-axis represents the FPR and the y-axis represents the TPR. With an AUC of 0.997, the model demonstrates excellent accuracy, indicating a strong ability to distinguish between relevant and irrelevant results with minimal errors in retrieval.

The variation of MAP with Epoch will be explained in Fig. 10. The MAP gradually increases to 1 as long as the Epoch is also enlarged from 1 to 40. Such a variable behavior indicates that the retrieval performance of the proposed model is also improved gradually and more positive samples with minimal false negatives can be retrieved effectively. The MAP value will be stable with an increase of Epoch if it is larger than 40, which indicates that the model training is essentially completed and the retrieval performance is also stable.

Figure 10 The MAP of the DHCNN on a multi-plant dataset with different epochs.

The MAP increases gradually from 1 to 40 epochs, indicating improved retrieval performance. Once the epoch exceeds 40, MAP stabilizes, suggesting that model training is complete and retrieval performance is stable.

Error analysis

In addition to the overall performance of the DHCNN model, some variations in retrieval accuracy were observed across different plant species datasets. Notably, in the single-species retrieval experiments discussed in ‘Search Results for Single Plant Disease’, DHCNN achieved an exceptionally high mean average precision of approximately 100% for the apple leaf disease dataset, outperforming its performance on the multi-species dataset. This superior result can be attributed to the distinct differences between the diseases in the apple dataset, which made it easier for the model to distinguish between classes, leading to near-perfect accuracy. In contrast, the corn leaf disease dataset posed more challenges, with DHCNN achieving a MAP of 98.4%, lower than its performance on the multi-species dataset. This can be explained by the significant intra-class similarities among the corn disease images, where certain disease symptoms are visually alike, resulting in a slight drop in retrieval accuracy. Similarly, the tomato leaf disease dataset demonstrated slightly lower retrieval accuracy, with a MAP of 99.4%, below the average. This lower performance can be attributed to the large number of disease categories within the tomato dataset. Diseases affecting the same species tend to have visual similarities, making it more challenging to differentiate between them compared to diseases across different plant species. This intra-class similarity, coupled with a greater number of disease categories, contributed to the reduced MAP for tomato disease retrieval. On the other hand, the retrieval results for diseases affecting grapes, potatoes and other plants were consistent with the performance on the multi-species dataset, showing no significant performance drops. These variations highlight the model’s sensitivity to different plant disease datasets. In the following sections, the performance of the DHCNN model will be evaluated from various aspects to provide a more comprehensive assessment of its overall effectiveness.

Ablation study

To evaluate the impact of the hash layer and different combinations of loss functions on the model’s performance, a series of ablation experiments were conducted. The results are summarized in Table 6. These experiments focus on different backbone networks, with or without a hash layer, examining how varying hash code lengths and loss functions affect precision, recall (TPR), and F-score.

Table 6 Impact of hash layer, hash code length and loss function on model performance.

Backbone network	With/without hash layer	Hash code length	Loss function	Precision	Recall (TPR)	F-score	
VGG-11	With Hash Layer	16	L3	0.993	0.994	0.993	
VGG-11	With Hash Layer	32	L3	0.995	0.996	0.995	
VGG-11	With Hash Layer	64	L3	0.997	0.997	0.997	
VGG-11	With Hash Layer	64	L1	0.135	0.124	0.129	
VGG-11	With Hash Layer	64	L2	0.939	0.926	0.932	
VGG-11	Without Hash Layer	–	L2	0.789	0.791	0.790	
Alexnet	With Hash Layer	64	L3	0.994	0.993	0.993	
resnet18	With Hash Layer	64	L3	0.997	0.996	0.997	

First, for the VGG-11 model, it is observed that performance improves consistently as the hash code length increases. When the hash code length is set to 64, using the L3 loss function (a weighted combination of cross-entropy loss and regularization loss), the model achieves the highest F-score of 0.997. This indicates that a longer hash code captures more discriminative information, thereby improving the model’s retrieval performance.

However, when using only the L1 loss (regularization loss), despite having a 64-bit hash code, the model’s performance drops significantly, with an F-score of only 0.129. This suggests that relying solely on regularization loss, which primarily optimizes Hamming distances between samples, is insufficient for effectively distinguishing between different categories, leading to poor retrieval results.

In contrast, when using the L2 loss (cross-entropy loss), the model’s performance improves significantly, achieving an F-score of 0.932. This shows the strength of cross-entropy loss in optimizing label prediction; however, it does not fully exploit the potential of the hash layer, indicating room for further improvement.

Notably, when the hash layer is removed, the F-score of VGG-11 drops to 0.790, confirming the necessity of the hash layer for enhancing retrieval performance.

Additionally, comparative experiments with AlexNet and ResNet18 as different backbone networks further validate the effectiveness of the L3 loss function. With the hash layer included, both AlexNet and ResNet18 achieve high performance, with F-scores of 0.993 and 0.997, respectively.

Overall, the experimental results demonstrate that the combination of the hash layer with the L3 loss function significantly enhances the model’s retrieval capability. In the next step, the performance of the proposed model will be further analyzed in comparison to other mainstream models across different datasets to validate its generalization ability.

Performance evaluation

The performance of two networks, AlexNet and VGG, fused with DHCNN will be compared. Both AlexNet and VGG are well-established models in deep convolutional neural networks. AlexNet offers several advantages, including a simpler network structure, fewer parameters and faster training speed, making it more suitable for straightforward image recognition tasks. On the other hand, VGG, with its deeper network structure, increased parameters, stronger feature representation capabilities and better transferability, is more adept at handling complex image recognition problems, despite the higher computational complexity it brings. To assess the retrieval performance of models using different backbone networks, AlexNet and VGG are employed as the backbone networks. Furthermore, three different hash code lengths, namely 16 bits, 32 bits and 64 bits, are selected to evaluate the experimental effects influenced by them. The MAP values for retrieval using different hash code lengths under the AlexNet and VGG backbone networks are presented in Table 7. The best retrieval performance is attained with the combination of the VGG backbone network and 64-bit hash codes, resulting in an average retrieval accuracy of 0.9974. When compared to the results of models proposed by Patil & Kumar (2017) and Piao et al. (2017), it is evident that the retrieval accuracy achieved in this study is significantly higher, even when dealing with more complex plant and disease categories, underscoring the effectiveness of the approach.

Table 7 Comparison of retrieval results between DHCNN and traditional retrieval methods.

Evaluation metrics	Retrieval using color, shape and texture features (De Boer et al., 2005)	DHCNN	
					Alexnet	VGG	
	LGGP	Hist	SIFT	LHS	16 bits	32 bits	64 bits	16 bits	32 bits	64 bits	
MAP@5	0.466	0.473	0.413	0.800	0.9919	0.9931	0.9938	0.9934	0.9954	0.9966	
MAP@10	0.343	0.383	0.296	0.720	0.9918	0.9930	0.9937	0.9932	0.9953	0.9965	
MAP	–	–	–	–	0.9927	0.9943	0.9952	0.9941	0.9964	0.9974	
AUC	–	–	–	–	0.9992	0.9993	0.9994	0.9993	0.9995	0.9997	
TPR	–	–	–	–	0.9913	0.9935	0.9957	0.9930	0.9954	0.9966	
FNR	–	–	–	–	0.0087	0.0065	0.0043	0.0070	0.0046	0.0034	

Finally, the retrieval precision, recall and F-score of the DHCNN model were comprehensively compared and analyzed with the latest methods in Table 8 (Karthikeyan & Raja, 2023; Patil & Kumar, 2013; Hussein, Mashohor & Saripan, 2011; Jadoon et al., 2017; Lu et al., 2021). The experimental results indicate that the PLDIR-CM and PSO-LIR models have lower values of precision, recall and F-scores. In contrast, the precision, recall and F-scores of the PNN, SVM and KNN models showed slight improvements. Additionally, the DWT-CBIR, CBIR-CSTF and DTLDN-CBIRA models achieved reasonable values for precision, recall and F-scores. However, the DHCNN technique outperformed the other methods, with the best precision, recall and F-scores being 99.5%, 99.7% and 99.59%, even when dealing with more complex plant and disease categories.

Table 8 Comparative analysis of DHCNN technique with recent methods.

Methods	Precision	Recall (TPR)	F1-Score	
PLDIR-CM	53.80	55.62	54.90	
DWT-CBIR	97.62	76.31	85.36	
PSO-LIR	19.17	69.63	34.82	
PNN Model	82.42	71.52	81.02	
SVM Model	88.79	78.65	86.17	
KNN Model	87.00	80.09	81.27	
CBIR-CSTF	96.00	80.75	86.55	
DTLDN-CBIRA  (Simonyan & Zisserman, 2014)	100.00	80.75	86.55	
DHCNN	99.50	99.66	99.58	

In order to mitigate the influence of a single dataset on the retrieval results of the DHCNN model, PlantDoc and the Plant Disease dataset have been additionally introduced. PlantDoc (Singh et al., 2020) is a dataset for visual plant disease detection. The dataset contains 2,598 data points in total across 13 plant species and up to 17 classes of diseases. Plant disease dataset (Chouhan et al., 2019): 12 economically and environmentally beneficial plants have been selected for this dataset. This dataset consists of 4,503 images, including 2,278 images of healthy leaves and 2,225 images of diseased leaves. In this study, the processing approach for the PlantDoc dataset and the Plant Disease dataset is consistent with the method described in ‘Proposed Methodology’ of this paper. The specific experimental results can be found in Table 9. It is evident that the DHCNN performs consistently well across different datasets.

Table 9 The retrieval accuracy of various datasets.

Plant leaf datasets	DHCNN	Other methods	
	MAP@5	MAP@10	MAP	MAP	
Plant-Village	0.9966	0.9965	0.9974	–	
PlantDoc	0.8212	0.8184	0.8204	0.7 (Patil & Kumar, 2017)	
Plant disease dataset	0.9709	0.9708	0.977	–	

As depicted in Fig. 11, the AP values under different hash code lengths and backbone networks consistently maintain high precision as k varies from 1 to 800, showcasing a robust and stable retrieval capability. However, the AP results decrease when k exceeds 800. This suggests that most of the relevant images in the retrieval dataset have already been retrieved. Furthermore, the retrieval performance of the model with the VGG backbone network surpasses that of the model with AlexNet. Additionally, increasing the number of hash bits gradually enhances the retrieval performance of the model. This demonstrates the effectiveness of utilizing a deeper network structure and a higher number of hash bits in improving the retrieval accuracy.

Figure 11 The AP at k (AP@k) of the DHCNN on a multi-plant dataset (Plant-Village) with different hash code bits and backbone network.

Conclusion

In this study, the DHCNN was introduced into the field of multi-plant leaf disease retrieval. The DHCNN integrates a pre-trained convolutional neural network, a hashing layer and a fully connected layer with a softmax classifier. This architecture effectively combines deep feature extraction with hash-based metric learning, transforming similar features into compact hash codes. The collision-resistant nature of hashing technology allows the model to distinguish fine-grained features in plant diseases, improving the ability to differentiate between highly similar disease characteristics. By integrating these refined hash codes with the class distribution information generated by the softmax classifier, the DHCNN significantly enhances the accuracy and efficiency of retrieving images depicting various plant leaf diseases.

To validate this approach, an augmented version of the PlantVillage dataset was utilized, demonstrating the effectiveness of the DHCNN in both single-plant and multi-plant leaf disease retrieval. The collision-resistant properties of the hashing technique played a crucial role in distinguishing highly similar disease features, with both accuracy and recall exceeding 98.4% in single-plant disease retrieval for crops like apple, corn and tomato. In multi-plant disease retrieval, the hashing technique further enhanced retrieval accuracy, achieving precision, recall and F-score values of 99.5%, 99.6% and 99.58% on the augmented PlantVillage dataset.

To further assess the model’s generalizability and stability, experiments were conducted on the PlantVillage, PlantDoc, and Plant Disease datasets. The model exhibited strong performance across these diverse datasets, confirming the robustness of the DHCNN. These findings reveal that the DHCNN holds considerable promise in addressing the complexities and variability inherent in plant disease retrieval. Overall, the DHCNN represents a significant advancement in plant disease retrieval, offering valuable insights for improving retrieval systems using deep learning and hashing techniques.

Future Directions

It is concluded from this study that the excellent performance in large-scale image retrieval for plant diseases based on the DHCNN has been achieved because of the high accuracy. Building on the success of our intelligent approach for plant disease retrieval, future research could explore the integration of more advanced deep learning techniques, to further enhance the accuracy and efficiency of disease retrieval. Additionally, expanding the method to include real-time applications and mobile-based platforms could make it more accessible to farmers and agricultural professionals in field conditions. Exploring the scalability of the model to include a wider variety of plant species and diseases, as well as investigating its applicability in diverse environmental conditions, would be valuable directions for future studies. These advancements could drive significant progress in precision agriculture, ultimately contributing to more sustainable and efficient farming practices.

Supplemental Information

Supplemental Information 1 Code

Additional Information and Declarations

Competing Interests

Author Contributions

Data Availability

The authors declare there are no competing interests.

Zhanpeng Yang conceived and designed the experiments, performed the experiments, analyzed the data, performed the computation work, prepared figures and/or tables, authored or reviewed drafts of the article, and approved the final draft.

Jun Wu conceived and designed the experiments, performed the experiments, analyzed the data, performed the computation work, prepared figures and/or tables, authored or reviewed drafts of the article, and approved the final draft.

Xianju Yuan conceived and designed the experiments, performed the experiments, analyzed the data, performed the computation work, prepared figures and/or tables, authored or reviewed drafts of the article, and approved the final draft.

Yaxiong Chen conceived and designed the experiments, analyzed the data, prepared figures and/or tables, authored or reviewed drafts of the article, and approved the final draft.

Yanxin Guo conceived and designed the experiments, prepared figures and/or tables, authored or reviewed drafts of the article, and approved the final draft.

The following information was supplied regarding data availability:

The data is available at Mendeley Data: Yang, Zhanpeng (2024), “Disease Retrieval Utilizing DHCNN”, Mendeley Data, V1, doi: 10.17632/v8kh23czrd.1.

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
