# Peer review of "General retrieval network model for multi-class plant leaf diseases based on hashing"

_PeerJ Computer Science, doi:10.7717/peerj-cs.2545_

## Round 0.1 · original submission · Major Revisions

Based on the referee reports, I recommend a major revision of the manuscript. The author should improve the manuscript, taking carefully into account the comments of the reviewers in the reports and resubmit the paper.

·

Basic reporting

Manuscript ID Submission ID 103159v1
This paper is related to reviewing the manuscript titled " General retrieval network model for multi-class plant leaf diseases based on hashing"
In their research, the authors introduce an innovative method employing Deep Hash Convolutional Neural Networks to overcome conventional obstacles and boost efficiency. Implementing Hash Learning has markedly increased the method's efficacy for large-scale data retrieval. The reliability of this retrieval technique has been confirmed using several well-known plant disease datasets.
Firstly, Although the proposed study is successful in terms of organization, presentation, content and results, major revision given in the following items need to be performed.

Experimental design

1) Provide the major numerical findings and conclusions of the study in the summary section another than accuracy performance criterion.
2) The proposed Hash-based deep architecture, CNN structure is given in Figure 1 and hash structure is given in Figure 2. However, how are deep CNN and hash structure combined and how is the mathematical model integrated? The paper should provide a solution to this problem and explain it in detail.
3) It would be better if the retrieval content of section 2.3 was transferred and combined with the previous section 2. Because section 2.3 is also related to the previous feature extraction section.
4) What is the relationship and/or difference between the L3 loss function in Equation 6 and the J loss function in Equation 7? Why are two different function equations defined?
5) The parameters and values given for the network model in Table 2 seem to be incomplete. In fact, this table should be expanded and a detailed parameter and values for the deep network model and hash structure should be presented.
6) Especially in the Introduction section and other sections of the article, current studies on deep neural networks, CNN, etc. for the years 2022-2024 should be used and these studies should be added to the reference section. Because, in fact, much older studies were used.
7) In the performance analysis, more performance analyses such as, AUC, TPR, FNR, except recall, precision and MAP, will increase the value and importance of the study.
8) Mentioning future studies and trends in the Conclusion section may be useful for readers.
9) In some parts of the article, the abbreviation "Deep Hash Convolutional Neural Networks" (DHCNN) is forgotten, other such abbreviations and explanations should be checked and reviewed.

Validity of the findings

As above

Additional comments

My decision is major revision. I do not see any harm in publishing the manuscript once the above revisions are made.

Reviewer 2 ·

Basic reporting

- This manuscript presents the General Retrieval Network Model for Multi-Class Plant Leaf Diseases.

- The manuscript is not well-organised and needs major revision.

- Literature should be completely remove from Introduction section. A new section should be for Related works.

- Contribution of this study should be highlighted as a separate sub-section.

- Proposed approach section is missing.

Experimental design

- F-score is not defined in section 4.1

- Precision and recall is not well. It should be defined in terms of True/False positive and True/False negative.

- Ablation study is missing.

Validity of the findings

- Novelty is unclear.

- Results are looking unrealistic. It touches 100% accuracy for 'Apple' in table-3. DHCNN is around 99% in terms of precision, recall, and f-score.

- Conclusion seems like a discussion. Not well concluded.

- Future works must be as a separate section.

Additional comments

Reject

Reviewer 3 ·

Basic reporting

All comments have been added in detail to the last section.

Experimental design

All comments have been added in detail to the last section.

Validity of the findings

All comments have been added in detail to the last section.

Additional comments

Review Report for PeerJ Computer Science
(General retrieval network model for multi-class plant leaf diseases based on hashing)

1. Within the scope of the study, a deep learning-based retrieval method was proposed using a dataset containing many different types of plant leaves.

2. In the introduction, the current status of approaches related to pest identification/prediction and plant disease, and the literature of artificial intelligence in this field are mentioned. In this section, the literature review needs to be organized. It is recommended to add a literature table consisting of sections such as "data type, data amount, data preprocessing/augmentation, problem type, results, evaluation metrics" to emphasize the importance of the subject and to summarize the literature clearly.

3. At the end of the first section, immediately after the literature review, the difference of the study from the literature, the main contributions to the literature and the originality points of the study should be explained in detail and clearly.

4. The first paragraph in the introduction is kept very short. It is recommended that this section be further detailed with sections such as plants, plant species, difficulties encountered, etc. to emphasize the subject.

5. The originality of the study should be clearly stated together with the algorithm and block diagram of the proposed model.

6. When figure-1 and related explanations are examined for feature extraction, it is observed that a VGG-based process is preferred. When the literature is examined, although there are many different CNN-based approaches that can be used in this section, please explain the reasons for preferring this in more detail.

7. The Plant-Village dataset used in the study and the details in table-1 are sufficient in terms of data amount, class type, dataset preference and data augmentation. However, since the obtained results are very dependent on the dataset distribution (training, testing), it should be explained how the distribution percentage is determined and whether different trials are performed.

8. For the evaluation metrics required for the analysis of the results, precision, recall, mean average precision and related graphs were obtained and it is observed that they are at a sufficient level within the scope of this study.

9. Parameter information such as learning rate and epoch used for the proposed deep hash CNN model is given. However, how these parameters are determined and/or preferred should be explained in detail. Since parameter changes and how they are determined may have a positive/negative effect on the results, this section should be detailed further.

10. Although the conclusion section is basically at a certain level, it is recommended that there be details regarding future works.

As a result, the study is important as a subject, but all the above-mentioned parts such as model selection, introduction, originality should be explained step by step and the relevant parts should definitely be updated in the paper.

---

## Round 0.2 · Minor Revisions

Kindly revise the manuscript as per the final reviewer suggestions and resubmit it.

·

Basic reporting

Manuscript ID Submission ID 103159v2 This paper is related to reviewing (For Round 2) the manuscript titled " General retrieval network model for multi-class plant leaf diseases based on hashing"
In their research, the authors introduce an innovative method employing Deep Hash Convolutional Neural Networks to overcome conventional obstacles and boost efficiency. Implementing Hash Learning has markedly increased the method's efficacy for large-scale data retrieval. The reliability of this retrieval technique has been confirmed using several well-known plant disease datasets.
First of all, in the second round of the proposed study, it seems that the authors did not complete and/or forgot some of the items in the first revision below. Therefore, the important revisions given in the following items must be reviewed and made again.

Experimental design

see below

Validity of the findings

see below

Additional comments

1. Revision 1: Provide the major numerical findings and conclusions of the study in the summary section another than accuracy performance criterion.
 No findings of any other performance metric other than the accuracy required in Revision 1 were included.
2. The proposed Hash-based deep architecture, CNN structure is given in Figure 1 and hash structure is given in Figure 2. However, how are deep CNN and hash structure combined and how is the mathematical model integrated? The paper should provide a solution to this problem and explain it in detail.
 In Revision 1, what was requested was not fully explained, the figures and equations in the previous version were repeated and the answer was given.
3. It would be better if the retrieval content of section 2.3 was transferred and combined with the previous section 2. Because section 2.3 is also related to the previous feature extraction section.
 Done.
4. What is the relationship and/or difference between the L3 loss function in Equation 6 and the J loss function in Equation 7? Why are two different function equations defined?
 Done.
5. The parameters and values given for the network model in Table 2 seem to be incomplete. In fact, this table should be expanded and a detailed parameter and values for the deep network model and hash structure should be presented.
 Done.
6. Especially in the Introduction section and other sections of the article, current studies on deep neural networks, CNN, etc. for the years 2022-2024 should be used and these studies should be added to the reference section. Because, in fact, much older studies were used.
 It is noticed that the current references for the years 2022-2024 requested in Revision 1 have not been added yet.
7. In the performance analysis, more performance analyses such as, AUC, TPR, FNR, except recall, precision and MAP, will increase the value and importance of the study.
 Done.
8. Mentioning future studies and trends in the Conclusion section may be useful for readers.
 Done.
9. In some parts of the article, the abbreviation "Deep Hash Convolutional Neural Networks" (DHCNN) is forgotten, other such abbreviations and explanations should be checked and reviewed.
 Done.

Reviewer 2 ·

Basic reporting

- Authors have addressed most of the comments, but still this manuscript needs improvement.

- Space is missing "2.Related Work". Check for all.

- Add a paragraph in the end of Section-1, which explains about the rest of the section in the manuscript.

- No need to mention 1.1 as Background. Remove this line.

- Expand introduction section more.

- Expand our contributions section in more details.

- Add a sub-section in Related work section for ML and DL papers.

Experimental design

- Add error analysis sub-section in the experimental sub-section.

- Replace F1 score to F-score for better visibility. Although, both are same.

- Add a sub-section called Ablation study for component-based analysis of this work.

Validity of the findings

- Add a sub-section called "Current status and Limitation" in Related work section wherein, highlight novelty in details as compared to previous studies.

- Add RoC-AUC graph in the manuscript.

- Conclusion and Future works are good now.

- Sections 3.2 and 3.3 should be the part of Experimental section.

- You are leveraging section - 3 for Proposed Methodology. It should be reflected in terms of section title also. So, arrange accordingly.

Additional comments

Need Minor revision.

Reviewer 3 ·

Basic reporting

All comments have been added in detail to the last section.

Experimental design

All comments have been added in detail to the last section.

Validity of the findings

All comments have been added in detail to the last section.

Additional comments

Review Report for PeerJ Computer Science
(General retrieval network model for multi-class plant leaf diseases based on hashing)

The responses given and the changes made to the paper are sufficient. I recommend that the paper be accepted as it is.

---

## Round 0.3 · Minor Revisions

Kindly address reviewer 1 comments properly and resubmit it.

·

Basic reporting

The authors still have not done what is requested in item 1. In fact, it is highlighted in red in the response letter to the referees, but this section has not been added to the abstract section of the paper as a follow-up to the change. If the authors do this as well, it would be appropriate for this article to be published.

Best Wishes

Experimental design

As above

Validity of the findings

As above

Additional comments

As above

Reviewer 2 ·

Basic reporting

Authors have addressed all comments.

Experimental design

- Authors have addressed all comments related to experimental works.


- Just replace F score to F-score in equation 16. (sub-section 4.2)

Validity of the findings

Authors have addressed all comments

Additional comments

Authors have addressed all comments. This manuscript can be accepted now.

---

## Round 0.4 · accepted · Accept

The Authors have addressed reviewer comments properly. Thus I recommend publication of the manuscript.